# In Vitro Assembly of Virus-Like Particles and Their Applications

**DOI:** 10.3390/life11040334

**Published:** 2021-04-10

**Authors:** Dinh To Le, Kristian M. Müller

**Affiliations:** Cellular and Molecular Biotechnology, Faculty of Technology, Bielefeld University, 33615 Bielefeld, Germany; dinh_to.le@uni-bielefeld.de

**Keywords:** capsid protein, in vitro assembly, virus-like particle, VLP-based vaccines, drug delivery

## Abstract

Virus-like particles (VLPs) are increasingly used for vaccine development and drug delivery. Assembly of VLPs from purified monomers in a chemically defined reaction is advantageous compared to in vivo assembly, because it avoids encapsidation of host-derived components and enables loading with added cargoes. This review provides an overview of ex cella VLP production methods focusing on capsid protein production, factors that impact the in vitro assembly, and approaches to characterize in vitro VLPs. The uses of in vitro produced VLPs as vaccines and for therapeutic delivery are also reported.

## 1. Introduction

Virus-like particles (VLPs) are highly structured protein complexes that resemble a native virus capsid. VLPs typically represent gene-less virus shells but in a wider definition encompass any type of capsid-derived nanoparticle. VLPs assemble from single or multiple structural capsid proteins inside appropriate production hosts, which is often termed in vivo assembly, as the host is seen as a single-cell organism, or in defined, cell-free conditions, also termed in vitro assembly. VLPs in general can be either enveloped or non-enveloped [1], whereby in vitro assembly typically leads to protein-only shells. Due to the lack of viral genomes, these particles are non-replicative but typically retain the transduction potential of the parental viruses, which may be used for nanoparticle delivery systems [2].

In vivo-produced VLPs have been exploited for different biomedical applications, such as vaccines, drug delivery and nanomaterials [3,4,5]. However, in this production strategy, capsid proteins are expressed and concomitantly assembled into VLPs inside the host cells, which may encapsidate host-related contaminations and thus impede VLP use. [6,7].

Along with progress in understanding virus assembly requirements, in vitro VLP production based on various viruses had been established. For the in vitro assembly, the capsid proteins are expressed and purified from expression hosts or cell-free protein synthesis systems (CFPS), or the proteins are obtained from in vivo VLPs via a disassembly procedure. This avoids trapping host-derived impurities, a potential source for unwanted immune response. Next, the capsid proteins are incubated under defined chemical conditions which promote VLP assembly. Compared to the in vivo VLPs, the in vitro technology offers the possibility to mix different capsid proteins or epitopes within a particle, for example resulting in a vaccine candidate for different viral genotypes [8,9]. For therapeutic delivery applications, the in vitro technology also enables the possibility to mix and match as well as control the amounts of different payloads during the capsid assembly reaction [4].

There are a few literature reviews on VLPs that focus on production [10] and various applications [3,4,5,11], but these do not discriminate between in vivo and in vitro VLPs. Here, we focus on in vitro produced VLPs covering the three domains: capsid-protein production, in vitro VLP formation and VLP characterization. We report in vitro VLP assembly protocols that have been applied in the context of vaccine development and drug delivery.

## 2. In Vitro Assembly of Virus-Like Particles (VLPs)

### 2.1. Capsid Protein Production

To prepare protein monomers for a cell-free capsid assembly under defined and controllable conditions, viral capsid protein can be obtained either by expression and purification of non-assembled proteins or via a disassembly procedure from in vivo generated and purified VLPs. The choice of the hosts for capsid protein expression significantly affects all further steps. *E. coli* is a preferred host for high capsid-protein expression [12]. Depending on the viral protein structure and expression method, proteins are either expressed in soluble form, which is normally favorable for capsid assembly, or as aggregates forming inclusion bodies within *E. coli*. To increase the soluble protein yield during expression, different factors such as the choice of the *E. coli* strain, expression vector, codon usage, expression temperature, induction condition and medium can be optimized [10,13]. Furthermore, tags have been fused to viral protein termini at the genetic level to aid protein folding and/or purification. Taking into account that the additional tag may interfere with capsid formation, a protease cleavage site can be added and used prior to assembly [14,15,16]. If proteins tend to form inclusion bodies in *E. coli* owing to incorrect folding, this expression strategy may be exploited with a strong promoter to produce high yields per dry mass and ease initial purification. The inclusion bodies can be easily separated from the cell debris and solubilized using strong denaturants like 6 M guanidinium HCl or 6–8 M urea [17,18,19,20,21].

Other favorable hosts for capsid protein production are Sf9 insect cells in combination with baculovirus vectors and yeast. Both organisms are amenable to scale up and offer the advantage of eukaryotic post-translational protein modification [22,23,24]. Lastly, cell-free protein synthesis systems have also been used to produce viral capsid proteins in defined transcription/translation reactions. This method allows for a direct control of capsid protein expression and VLP assembly conditions and enables production of toxic and insoluble proteins [25,26,27,28].

If VLPs assemble spontaneously inside the expression hosts, they may contain undesired compounds. To obtain protein subunits from VLPs formed inside the host cell, different disassembly methods exist. Bacteriophage MS2 VLPs produced in *E. coli* were disassembled at low pH, while human papillomavirus (HPV) VLPs obtained from insect cells were dissociated with carbonate buffer at pH 9.6 [29,30]. Cowpea chlorotic mottle virus (CCMV) particles assembled in plants were denatured in a neutral buffer with high salt concentration [31,32,33]. Urea is commonly used to disassemble the formed VLPs. Bacteriophage Qβ VLPs were disassembled in 6 M urea [34], whereas 2.5 M urea were sufficient to denature hepatitis B core protein (HBc) VLPs [35]. Reducing and chelating agents can also be used in VLP disassembly [30,36,37,38,39,40,41].

### 2.2. Factors Impacting In Vitro Assembly

The capsid self-assembly is driven by Brownian motion and interactions between subunits as well as subunits and other viral or non-viral components to minimize free energy in higher structures. The capsid assembly is proposed to occur in three steps. First, a capsid oligomer nucleus is formed from capsid proteins in the so-called nucleation phase. Afterwards, building blocks (a protein monomer, or capsid oligomers) are added to the nucleus during the growth phase. Finally, the last building block is inserted to complete the capsid [42,43,44]. The assembly process is complicated and highly dependent on the viral protein structure and the experimental conditions. Conceptually, VLP formation in the first place is governed by general rules applicable to protein folding and stability while considering aggregation [45]. We present here the main factors that affect in vitro assembly (Figure 1).

#### 2.2.1. pH, Ionic Strength, and Temperature

pH plays an important role for in vitro assembly since it affects the capsid-protein charge. Most in-vitro assembly reactions were optimized at a physiological pH [15,18,46]. Interestingly, some particles assemble at acidic or alkaline pH. In vitro assembly studies of CCMV showed that the attraction between capsid proteins was optimal at about pH 5 and decreased sharply with increasing pH [47]. Rotavirus (RV) VLPs formed in vitro at an acidic pH [21]. Our in vitro assembly results from adeno-associated virus serotype 2 (AAV2), however, indicated that an alkaline pH (pH 9) was most suitable to avoid aggregates and promote VLP formation [19].

Ionic strength is another main factor for capsid assembly in vitro. Salts interact with charges on the protein surface, influence the water shell, disfavor hydrophobic exposure and ultimately affect protein stability [48]. To optimize in vitro capsid-assembly reactions, ionic strength needs to be optimized along with the change of pH [15,18,46], which can be presented as phase diagrams of the protein assembly [49].

The effect of temperature on the capsid assembly also deserves evaluation. Low temperatures are normally favorable as they reduce protein aggregation and chemical degradation. For example, the in vitro assembly of primate erythroparvovirus 1 (B19) at 4 °C showed a better yield at 4 °C than at 37 °C [18]. Contrarily, near-physiological temperatures promoted the Rous Sarcoma virus capsid formation in vitro [50]. Another report on hepatitis virus assembly revealed that subunit exchange with assembled capsid shells was generally slow but slightly elevated at lower temperature [51].

#### 2.2.2. Nucleic Acids, Scaffolding Protein, and Additives

The in vitro assembly of some capsids was highly influenced by the presence of nucleic acids. CCMV, a single-stranded RNA virus, is a well-studied model. Garmann et al. demonstrated that the assembly of CCMV depends on balanced capsid protein (CP)–CP interactions relative to CP–RNA interactions [52]. Furthermore, a high CP/RNA mass ratio is required to assemble CCMV VLPs [53]. Recently, CCMV capsid proteins were shown to encapsidate both single-stranded DNA (ssDNA) in typical spherical assemblies and double-stranded DNA (dsDNA) in rod-like VLPs [54]. Other RNA viruses also require nucleic acids for in vitro assembly. The assembly of Gag proteins into human immunodeficiency virus (HIV1) VLPs in a defined system was directly supported by RNA [55]. Similarly, the in-vitro assembly of hepatitis C virus (HCV) nucleocapsid-like particles required structured RNA as reported by Kunkel et al. [56]. Bacteriophage MS2 was also assembled in vitro in the presence of nucleic acids [29]. The assembly of beak and feather disease virus (BFDV), a member of the circular ssDNA circovirus family, is regulated by its ssDNA genomes. The highly positive charged N-terminal arginine-rich motif of capsid proteins interacted with ssDNA during in vitro capsid assembly [57]. VLPs derived from simian virus 40 (SV40), a dsDNA virus, also needed DNA for efficient in vitro assembly [58]. Nonetheless, it should be considered that the additional nucleic acids within the in vitro VLPs might interfere with VLP applications. For example, nucleic acid can modulate the immune response by activating pattern recognition receptor (PRR) [6].

Scaffolding proteins are not a component of a mature capsid. These proteins assist during capsid formation. Providing a scaffolding protein to an in vitro capsid assembly in many cases increases the yield of fully assembled capsids. Our data on in vitro assembly of AAV2 showed that the addition of assembly-activating protein (AAP, an AAV scaffolding protein) helped to improve AAV2 capsid formation [19]. The assembly of different bacteriophages was highly dependent on scaffolding proteins, which promoted the polymerization of the major capsid protein [59,60,61,62,63]. The herpes simplex virus procapsid assembly reaction from purified major capsid proteins has been reported by Newcomb et al. to require a scaffolding protein and to form small procapsids at low concentrations of the scaffolding protein [64].

Small molecule additives can be used to aid capsid assembly. L-arginine improved the solubility of assembled VLPs by preventing aggregation during protein refolding [19,65]. In a study performed by Lampel et al., chemical chaperones, such as methylamines enhanced HIV-1 in vitro assembly [66]. Other reagents can be combined with capsid proteins during in vitro assembly leading to new hybrid materials. For example, CCMV capsid proteins have been explored in combination with different supramolecular templates. Organo Pt (II) complexes formed spherical or rod-like structures that were combined with the capsid proteins to form likewise shaped CCMV VLPs [67]. Micelles and DNA micelles were packaged inside CCMV VLPs that offer new drug-delivery systems, especially for hydrophobic drugs [68,69]. Polymers were also incorporated into CCMV VLPs during in vitro assembly [70,71].

### 2.3. VLP Characterization

To analyze the formed in vitro VLPs, different biochemical and biophysical methods can be applied. Standard methods are sodium dodecyl sulfate-polyacrylamide gel electrophoresis (SDS-PAGE) to determine size and purity, Western blot to confirm the identity, and ultraviolet (UV) spectroscopy with light scattering compensation, which monitors amino acids (phenylalanine, tyrosine and tryptophan) and nucleotides with an absorbance spectrum of about 240–300 nm, to determine concentrations [36,72,73]. Other protein quantitation methods, such as the bicinchoninic acid (BCA) assay and Bradford assay have been also used [18,36,74]. Size exclusion chromatography and dynamic light scattering (DLS) are widely used to characterize the size of VLPs [19,75,76,77]. The latter method provides the heterogeneity of the VLP samples and the mean hydrodynamic diameter of particles, which is normally greater than the physical diameter. Stability has been assessed using techniques such as differential scanning fluorimetry [78]. Structural integrity has been verified by circular dichroism spectroscopy (CD) and Fourier-transform infrared (FT-IR) spectroscopy [79]. The nucleic–capsid protein interactions have been determined by gel retardation assay, optical tweezers (OT) and acoustic force spectroscopy (AFS) [53,58]. VLP morphology and possible intermediate aggregates formed during an in-vitro assembly reaction can be visualized using transmission electron microscopy (TEM) or atomic force microscopy (AFM). TEM can also be used to distinguish between empty and encapsulated VLPs [52,59] and AFM can assess the VLP height profile [19,80,81]. A VLP high-resolution structure can be determined using Cryo-EM [52] or crystallography [82]. Mass spectrometry has been used to identify VLP composition [76]. If available, an assembled particle antibody helps to confirm VLP conformation in enzyme-linked immunosorbent assays (ELISAs) [19].

Understanding capsid self-assembly pathways will help to tailor the mechanisms of virus infection and replication for therapeutic applications. Different methods have been used to characterize intermediate assemblies and the assembly pathways of virus capsids, such as electron microscopy [83,84,85,86], X-ray crystallography [57], atomic force microscopy [57,58,85], small-angle X-ray scattering [87,88,89,90,91,92], mass spectrometry [93,94,95], size-exclusion chromatography [84], resistive-pulse sensing [84,96], interferometric scattering microscopy [97], single-molecule fluorescence correlation spectroscopy [98], optical tweezers in combination with confocal fluorescence microscopy and acoustic force spectroscopy [58,99]. Recently, high-speed atomic force microscopy (HS-AFM), a powerful single-molecule technique for real-time visualization of biomolecules in dynamic action [100], has been used to visualize self-assembly of HIV capsid protein lattice [81]. This physical virology technique will enable real-time capsid assembly studies of other viruses in the future.

## 3. Application of In Vitro-Assembled VLPs

### 3.1. Vaccine Development

VLPs are composed of protein monomer units in a repetitively ordered structure with a size range of 20–200 nm in diameter that is appropriate for vaccination [11]. VLPs are able to induce a strong immune response, which was described by Jennings et al. and Mohsen et al. [6,101]. A VLP derived from a pathogenic virus may be used to elicit an immune response directly to the parental virus, or function as a scaffold to present heterologous epitopes. Compared to subunit peptides or proteins, VLPs provide the possibility to present epitopes in a natural conformation that benefits an anti-viral B- and T-cell immune response. Moreover, due to the highly repetitive epitope presentation, VLPs can induce a strong B cell response even without adjuvants [102,103,104]. Another advantage is that a foreign epitope can be chemically or genetically incorporated onto the VLP surface, offering a flexible platform to create different vaccine candidates [105,106,107,108].

A few VLP-based vaccines have a long and very successful tradition in the clinic and recently several new candidates progressed to the clinic. These vaccine candidates have been produced in vivo or by an in vitro system [3]. Compared to in vivo VLP production, which potentially may be contaminated with host-derived components resulting in unpredictable immune responses that require an additional quality control effort [6], a cell-free VLP technology offers better control during production. A list of developed and approved vaccine candidates is presented in Table 1.

Human papillomavirus (HPV) is a major cause of cervical cancer and associates with many human diseases [127]. HPV capsids are composed of two capsid proteins, L1 and L2, assembled in a T = 7 icosahedral structure (about 60 nm in diameter) [128]. L1 is a structural protein and able to form HPV VLPs [129]. To date, there are four HPV vaccines on the market, which use L1-VLP platforms [3].

The HPV vaccine Cervarix, which is manufactured by GlaxoSmithKline and partially based on technology from MedImmune/AstraZeneca, was approved in 2007 in the European Union (EU) and other countries and 2009 in the USA. It contains two monovalent antigen bulks of C-terminally truncated versions of the major capsid proteins L1 of either serotype 16 or 18. The proteins are produced in cells derived from the *Trichoplusia ni* (Hi-5) insect cell line with recombinant baculoviruses encoding the L1 proteins. The proteins are released by osmotic shock and upon a multi-step purification by filtration and chromatography the proteins assemble into spherical particles, which are mixed and formulated with adjuvant. The final product dose contains 20 µg HPV 16 L1 protein, 20 µg HPV 18 L1 protein, 500 µg aluminium hydroxide, 50 µg 3-O-desacyl-4′ -monophosphoryl lipid A (MPL) and 0.624 mg NaH_2_PO_4_·2H_2_O [109].

The HPV vaccine Gardasil, which is produced by Merck Sharp and Dohme and was approved in the USA in 2010, contains the HPV L1 protein from serotypes 6, 11, 16, and 18. The vaccine Gardasil 9 (9-valent vaccine), which was approved in the USA in 2014, additionally contains the HPV L1 proteins of serotypes of 31, 33, 45, 52, and 58. The proteins are individually produced in the yeast from *Saccharomyces cerevisiae* transformed with the pGAL110 expression vector coding for the respective proteins. Cells are harvested by filtration, frozen and then VLPs are released by homogenization. For some L1 types (31, 33, 45, 52, 58), a protease inhibitor is added. Purification comprises cross-flow membrane filtration for debris removal, cation exchange chromatography for host cell protein removal, and hydoxyapatite chromatography for polishing and enrichment of monodisperse VLPs. For all but type L1 serotype 18, the VLPs are disassembled using dithiothreitol (DTT) and reassembled by removing DTT, which improves VLP structure and stability. A final buffer exchange yields the final aqueous products which are then adsorbed onto the adjuvant amorphous aluminium hydroxyphosphate sulfate by in-line mixing. The individual products are then mixed by sequentially adding them to a tank with buffer and alum adjuvant, settling and decanting. The final 0.5 mL aqueous dose Gardasil 9 contains 30 µg L1 6, 40 µg L1 11, 60 µg L1 16, 40 µg L1 18, and 20 µg of L1 31, 33, 45, 52, 58 as well as 500 µg aluminium (adjuvant as amorphous aluminium hydroxyphosphate sulfate), 9.56 mg NaCl (for stability), 0.78 mg L-histidine (for buffering), 50 µg polysorbate 80 (for VLP stability and prevention of aggregation or surface adsorption), 35 µg Na-borate (for buffering) [79,110].

As a biosimilar, the company Innovax produced Cecolin, a bivalent HPV16 and HPV18 vaccine, by expressing the respective L1 proteins in *E. coli* and maintaining them in soluble pentamer form under a reducing condition (20 mM DTT). The HPV VLPs were formed by removal of the reductant during protein purification [8]. The vaccine production in *E. coli* offers a low manufacturing cost and easy scale-up compared to other licensed products [111]. Similarly, Innovax produced an HPV9-valent vaccine candidate (HPV 6/11/16/18/31/33/45/52 and 58), which is currently in phase 2 clinical trial [9]. As a result of protein expression in *E. coli*, lipopolysaccharide (LPS) might contaminate into VLPs that can enhance immune responses and cause pyrogenic and shock reactions in mammals. To reduce LPS contamination, different methods, such as size exclusion chromatography, affinity chromatography, binding to polymyxin and treatment with Triton X-114 have been used [130].

Hepatitis E virus (HEV), which can be quasi-enveloped, is the most acute hepatitis cause in both developing and developed countries [131,132]. A 7.5-kb HEV genome comprises three overlapping open reading frames encapsidated into a T = 3 symmetry capsid of 27–34 nm in diameter [133,134]. The open reading frame 2 (ORF2) codes for viral capsid proteins, which were produced in *E. coli* or insect cells as recombinant HEV vaccine candidates [135]. Hecolin, a licensed HEV vaccine in China, is an HEV cell-free assembly product [113]. p239 truncated capsid protein (239 amino acid (a.a.) of a full-length capsid protein with 660 a.a.) was obtained from *E. coli* and assembled into VLPs of 20–30 nm diameter via multiple purification steps [78,112]. Another HEV vaccine candidate is currently under preclinical evaluation using p495 protein (495 a.a. truncated capsid protein) expressed in *E. coli* to form in vitro HEV VLPs [115].

Also surface antigens from enveloped viruses have been used in mostly defined assembly reactions. The surface antigen of the hepatitis B virus (HBV) which causes chronic infection of about 3.5% world population [136] named HBsAg was recombinantly produced in yeast. In a first version, the S gene of HBV was cloned in a plasmid and expressed in *Saccharomyces cerevisiae* strain DC5. Purification steps comprised cell disruption, diafiltration, size exclusion chromatography, ion-exchange chromatography, a CsCl ultracentrifugation followed by a final size exclusion chromatography. In the absence of chemical treatment, HBsAg formed spheres of about 20–22 nm containing non-glycosylated HBsAg and a lipid matrix consisting mainly of phospholipids [116,117]. This vaccine is marketed by GlaxoSmithKline as ENGERIX B, which was approved in 1986 in the EU as the first recombinant vaccine, and with added MPL adjuvant as Fendrix, which was approved in 2005. A similar product also produced in yeast is Recombivax HB from Merck [118,119]. The malaria vaccine Mosquirix from GlaxoSmithKline, which is also named RTS,S/AS01 and was approved in the EU in 2015, is produced by coexpression in yeast of the HBsAg with a fusion protein of HBsAg with the pre-erythrocytic circumsporozoite protein (CSP) of the *Plasmodium falciparum* malaria parasite. In this case, the mixed particles formed already during expression [137]. Hence, for these multi-subunit particles formed by surface antigens of enveloped viruses, which do not resemble typical capsids but defined aggregates, the differentiation between in-cell versus in-buffer VLP assembly is fluid. Compared to in vivo VLPs, the in vitro manufacturing process is defined by additional steps to reassemble VLPs outside the cells. In some cases, a dedicated assembly step is missing, because the in vitro VLP assembly occurs during steps ascribed to purification (Assembly method, Table 1). Such combined purification and assembly may reduce downstream processing costs. In other cases, few mutations may shift assembly between in vivo and in vitro.

Recently, the company Novavax developed the NVX-CoV2373 vaccine candidate (currently in phase 3 clinical trial) for preventing severe acute respiratory syndrome coronavirus 2 (SARS-CoV-2) infection, the virus which spread from China around the globe in 2020 causing the coronavirus disease 2019 (COVID-19) pandemic with millions of infections and deaths [120,138]. The SARS-CoV-2 spike protein (S protein), which is responsible for receptor binding and virus entry [139] was expressed in Sf9 insect cells using a baculovirus expression system. The protein was then obtained from the plasma membranes with a buffer containing NP-9 detergent. The 27.2 nm nanoparticles, potentially S-trimers anchored within polysorbate 80 (PS80) detergent cores, formed during protein purification and, concomitantly, detergent removal [77]. This strategy was also used to develop vaccine candidates against other coronaviruses (SARS, Middle East respiratory syndrome (MERS)) [121] and other pathogens including Influenza virus [122,123] and respiratory syncytial virus (RSV) [23,124,125].

Other vaccine candidates with different in-vitro assembly conditions are under preclinical evaluation. The C-terminally truncated hepatitis B core antigen (HBcAg) was produced using *E. coli*-based cell-free protein synthesis reaction and assembled into HBV VLPs [27]. The primate erythroparvovirus 1 (B19) VLPs presenting the peptides derived from F proteins of RSV virus were tested as an RSV vaccine candidate. The chimeric proteins were expressed in inclusion bodies in *E. coli*, and the VLPs were formed by dialyzing of the proteins from a denaturing buffer (5 M GuHCl) to PBS buffer [126]. A vaccine candidate for canine parvovirus (CPV) disease was also produced via an in vitro cell-free reaction. Xu et al. reported that CPV capsid protein (VP2) fused to a SUMO-tag was acquired from *E. coli* in soluble form. The CPV VLPs were developed during SUMO tag removal and dialysis of the capsid proteins into a physiological buffer [15].

### 3.2. Therapeutic Delivery

VLPs are attractive candidates to deliver drugs, small molecules or nucleic acids due to their biocompatibility, biodegradability and targeted delivery [4,140]. Different cargo-loading strategies have been explored with both in vivo and in vitro VLPs [141]. Here, we focus on different in vitro packaging approaches, which occur via disassembly/reassembly of VLPs or during in vitro assembly of purified proteins into VLPs. The in vitro encapsulation mainly relies on the interactions between loading cargoes and viral capsid proteins and differs among VLPs (Table 2). With small molecules not interfering with the assembly, concentration-dependent stochastic loading by engulfment during assembly is also feasible [142].

During protein expression, some capsid proteins tend to assemble VLPs inside the hosts. To remove potential host-related impurities and encapsulate cargoes, the disassembly and reassembly of VLPs are needed. The use of bacteriophage MS2 VLPs in cargo delivery was reported by different groups. Generally, MS2 capsid proteins were produced and formed 27.5 nm MS2 VLPs in *E. coli*. The VLPs were subsequently purified and disassembled in glacial acetic acid. Since MS2 VLPs are able to reassemble and encapsulate negatively charged cargoes in vitro at neutral pH, drugs, proteins or siRNA were loaded into the capsids for therapeutic delivery [29,76,143]. Targeting peptides were covalently linked to the capsid surfaces to specifically deliver the drugs to cells presenting the cognate receptor [29,143]. With a similar approach, Douglas et al. described a method to package a protein inside bacteriophage P22, which was performed by mixing of the cargo protein fused to a scaffold protein and bacteriophage P22 capsid proteins in a mild denaturing condition buffer (1.5 M GuHCl), followed by dialysis against a neutral buffer [144,145]. Another bacteriophage, Qβ has been used to encapsidate a fluorescent protein [34]. Hepatitis B core protein (HBc) VLPs were also exploited for drug delivery. The HBc VLPs assembled in *E. coli*, were denatured with urea, and reassembly and drug encapsulation was achieved by dialysis of the denatured proteins with a neutral buffer. The RGD peptide was genetically incorporated to HBc VLPs to target tumors [35]. In vitro VLPs were also explored in gene delivery. SV40 and HPV VLPs were produced in insect or mammalian cells. The particles were disassembled in the presence of reducing and chelating agents, DNA plasmids were then packaged inside the VLPs and delivered to target cells [37,38,39,41,146,147]. Human polyomavirus 2 (JC) VLPs produced in yeast have been used to deliver RNAi, which was loaded via an osmotic shock [148].

CCMV VLPs are widely tested for therapeutic delivery due to their pH-dependent capsid assembly [155]. CCMV VLPs were produced in plant cells, the particles were then disassembled in a high-salt concentration buffer at neutral pH. Reassembly and packaging followed by adding cargoes and dialyzing into an acidic buffer (at pH 4.5–4.8) [31,32,33], which can be used to package RNA replicons [156], or a neutral buffer which has been used with DNA cargoes [54]. CCMV VLPs were also produced using *E. coli* by expressing soluble capsid proteins, followed by purification and assembly in the presence of cargoes. To target subcutaneous cancers, folic acid (FA) was conjugated to CCMV capsid proteins [32,150]. Other VLPs obtained by in-vitro assembly were also tested for loading and drug delivery. Murine polyomavirus (MPyV) capsomeres fused to a desired protein were expressed in *E. coli* and the purified capsomeres were dialyzed resulting in the formation of VLPs containing the guest protein for delivery [75]. Zhao et al. described a method to deliver the chemotherapy drug doxorubicin (DOX) using rotavirus (RV). RV structural protein VP6 formed inclusion bodies during expression in *E. coli*, and the protein was then purified under denaturing condition (8 M urea). DOX was conjugated to denatured VP proteins and the assembly of VLPs occurred during protein dialysis into a low pH buffer [21].

For diagnostic purposes such as magnetic resonance imaging (MRI), magnetic nanoparticles can also be coated with capsid proteins and additionally endowed with a targeting function. VP1^ΔC589^, VP1^wt^ or VP1^N138C^ of SV40 were produced in Sf9 cells and pentamers were purified and assembled around magnetic nano-particles in MOPS (3-Morpholinopropane-1-sulfonic acid) buffer. The latter two particles were crosslinked with epidermal growth factor (EGF) for the targeting of EGFR expressing cells using heterobifunctional crosslinkers with N-hydoxysuccinimide and maleimide groups [151]. Similarly, SV40 VLPs produced in *E. coli* have been used to encapsulate quantum dots for imaging [152]. Brome mosaic virus (BMV) and Ross River virus (RRV) particles were also explored for medical imaging. The purified capsid proteins were mixed with functionalized gold nanoparticles (GNPs) and dialyzed against an assembly buffer. The encapsulation was regulated by the electrostatic interactions between capsid proteins and gold nanoparticles, followed by the capsid protein–capsid protein interactions [153,154].

## 4. Conclusions

VLPs have been widely exploited for vaccine development and therapeutic delivery with success in the clinic and promising preclinical evaluations. In vitro VLP production technology has emerged as a versatile technology besides the in vivo method. The formation of VLPs under controllable and defined conditions enables the technology to create multi-vaccine candidates by combining different antigens within a particle while avoiding the unpredictable immune response related to possible host-related contaminations. In vitro assembly of VLPs also offers a feasible tool to control the packaging amount and cargo components for therapeutic delivery.

In vitro VLP assembly is a complex process and differs among viral capsids. Even though many in-vitro VLPs have been produced, it is still unclear whether all virus capsids are amenable to ex cella production. Many factors need to be optimized for each candidate to assemble VLPs in vitro, improve assembly yields, and enable large-scale production. Cell-free protein synthesis in combination with in vitro VLP assembly bears potential for the generation of various synthetic biology products at a smaller scale. In vitro VLPs have been widely used as VLP-based protein vaccines. When in vitro nucleic acid encapsidation becomes more accessible, in vitro VLPs might also become an efficient and safe technology for vector VLP vaccines, which deliver an antigen-coding sequence, or for gene therapy vectors for treating diseases. Also for other therapeutic delivery strategies, recent advances provided mostly a proof of concept. Further work is needed to fully tap the potential of in vitro VLPs, but the future looks bright.

## Figures and Tables

**Figure 1 life-11-00334-f001:**
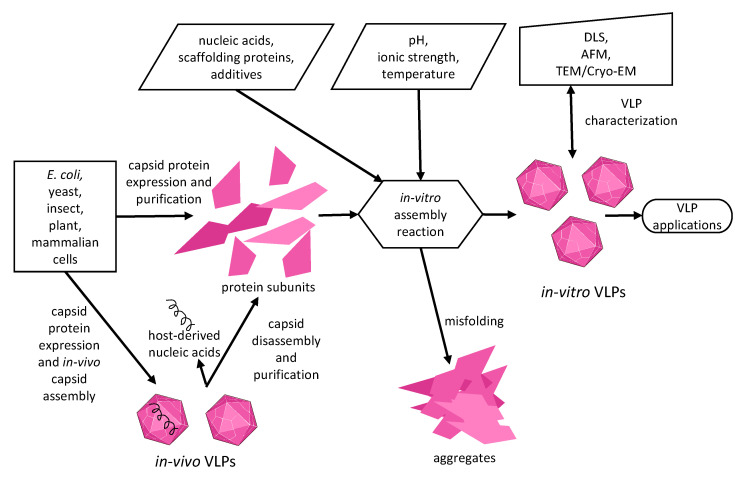
Schematic representation of in vitro virus-like particle (VLP) assembly. Capsid protein production methods, factors impacting on VLP in vitro assembly and typical methods to characterize VLPs are presented.

**Table 1 life-11-00334-t001:** In vitro produced VLP-based vaccines on the market or in development.

Vaccine Candidate	Host	VLP ^a^ Platform	Vaccine Antigen Proteins	Assembly Method	Development Stage	Ref.
Cervarix (GlaxoSmithKline)	Insect cell	HPV-L1	HPV 16 and 18 L1	Multi-step purification	Approved	[109]
Gardasil (Merck Sharp and Dohme)	Yeast	HPV-L1	HPV 6/11/16/18 L1	VLP disassembly using DTT/Reassembly by DTT removal	Approved	[79]
Gardasil-9 (Merck Sharp and Dohme)	Yeast	HPV L1	HPV 6/11/16/18/31/33/45/52/58 L1	VLP disassembly using DTT/Reassembly by DTT removal	Approved	[110]
Cecolin^®^ (Innovax)	*E. coli*	HPV L1	HPV16 and 18 L1	Protein purification and reducing agent removal	Approved (China)	[8,111]
HPV 9-valent (Innovax)	*E. coli*	HPV L1	HPV 6/11/16/18/31/33/45/52/58 L1	Protein purification and reducing agent removal	Phase 2 NCT03935204	[9]
Hecolin^®^ (Innovax)	*E. coli*	HEV p239	HEV truncated E2, 239 a.a.	Multi-step purification	Approved (China)	[78,112,113,114]
HEV	*E. coli*	HEV p495	HEV E2, 495 a.a.	Protein was dialyzed into 50 mM phosphate buffer with 0.5 M NaCl, pH 6.5	Preclinical evaluation	[115]
ENGERIX-B/Fendrix (GlaxoSmithKline)	Yeast	HBsAg	HBV HBsAg	Multi-step purification	Approved	[116,117]
Recombivax (Merck Sharp and Dohme)	Yeast	HBsAg	HBV HBsAg	Multi-step purification	Approved	[118,119]
Hepatitis B	Cell-free synthesis system	HBcAg	HBV truncated HBcAg	30 µL CFPS product was dialyzed against 100 mM HEPES and 200 mM NaCl, pH 7 1 mL CFPS product was dialyzed against 10 mM BisTris and 0.385 M NaCl, pH 5.5	Preclinical evaluation	[27]
SARS-CoV-2 (NVX-CoV2373, Novavax)	Sf9 cell	S-trimer nanoparticle	SARS-CoV-2 Spike	Removal of the detergent (Tergitol^TM^ NP-9) during protein purification	Phase 3 NCT04611802	[77,120]
MERS-CoV and SARS-CoV (Novavax)	Sf9 cell	S nanoparticle	MERS-CoV/SARS-CoV Spike	Removal of the detergent (Tergitol^TM^ NP-9) during protein purification	Preclinical evaluation	[121]
Influenza (NanoFlu^TM^,Novavax)	Sf9 cell	HA nanoparticle	Influenza virus HA	Removal of the detergent (TergitolTM NP-9) during protein purification	Phase 1/2 NCT04120194	[122,123]
RSV (Novavax)	Sf9 cell	F nanoparticle	RSV F	Removal of the detergent (TergitolTM NP-9) during protein purification	Phase 1 NCT03026348	[23,124,125]
RSV	*E. coli*	B19 VP2	Two peptides derived from RSV F	Denatured proteins in 5 M GuHCl were dialyzed into PBS buffer at 4 °C for 36 h	Preclinical evaluation	[126]
CPV	*E. coli*	CPV VP2	CPV VP2	Fusion proteins (SUMO tag fused to capsid protein) were cleaved by SUMO protease and dialyzed into 50 mM Tris-HCl, 150 mM NaCl, pH 7	Preclinical evaluation	[15]

^a^ B19: Erythroparvovirus 1; CPV: Canine parvovirus; HBcAg: Hepatitis B core antigen; HBsAg: the surface antigen of the Hepatitis B virus; HBV: Hepatitis B virus; HEV: Hepatitis E Virus; HPV: Human papillomavirus; MERS: Middle East respiratory syndrome coronavirus; RSV: Respiratory syncytial virus; SARS: Severe acute respiratory syndrome coronavirus; SARS-CoV-2: Severe acute respiratory syndrome coronavirus type 2; HEPES: 2-[4-(2-hydroxyethyl)piperazin-1-yl]ethanesulfonic acid.

**Table 2 life-11-00334-t002:** In vitro VLP delivery platforms.

VLP ^a^ Platform	Expression System	Cargo and Loading Method	Targeting	Assembly Method	Ref.
MS2	*E. coli*	Chemotherapeutic drugs (DOX), siRNA cocktails, protein toxins Conjugated to RNA	Hepatocellular carcinoma using SP94 peptide	VLP disassembly in glacial acetic acid/Reassembly and packaging: capsid proteins in 10 mM acetic acid, 50 mM NaCl, pH 4 were incubated with RNA in 50 mM Tris-HCl pH 8.5 buffer	[29]
MS2	*E. coli*	siRNA Genetic fusion to TR step loop	HeLa cells using human transferrin	VLP disassembly in glacial acetic acid/Reassembly and packaging in 40 mM ammonium acetate, pH 6 buffer	[143]
MS2	*E. coli*	Alkaline phosphatase Electrostatic interaction to capsid protein	-	VLP disassembly in glacial acetic acid/Reassembly and protein packaging in 50 mM Tris, 100 mM NaCl, 250 mM Trimethylamine *N*-oxide buffer	[76]
P22	*E. coli*	Streptavidin, Ferritin cages, CelB Genetic fusion to a scaffold protein (SP)	-	VLP disassembly in 3 M GuHCl/Reassembly and protein packaging by adjusting the mixture of coat proteins (CP) and fusion proteins to 1.5 M GuHCl, and dialyzing to the buffer of 50 mM Tris-HCl, 25 mM NaCl, 2 mM EDTA, 3 mM β-mercaptoethanol, 1% glycerol	[144,145]
Qβ	*E. coli*	Fluorescent protein (GFP)	-	Qβ VLPs disassembly in 20 mM Tris-HCl, 50 mM NaCl, 6 M urea, 10 mM DTT, and dialyzed against 10 mM acetic acid and 50 mM NaCl/Reassembly and GFP packaging in 50 mM NaCl, 20 mM Tris-HCl, pH 7.5	[34]
HBc	*E. coli*	DOX Insertion of hydrophobic peptide to capsid protein to confine DOX	Tumor-targeting peptide RGD	VLP disassembly in 2.5 M urea, 150 mM NaCl, 50 mM Tris-HCl/Reassembly and DOX packaging in the buffer of 50 mM Tris-HCl, 150 mM NaCl, 10% glycerol, 1% glycine, pH 8	[35]
SV40	Sf9	DNA plasmid (up to 17.7 kb) Interaction between capsid protein and dsDNA	-	VLP disassembly in the presence of DTT, EDTA, EGTA/Reassembly and DNA packaging in the buffer containing MgCl_2_, CaCl_2_ (ATP)	[41,146,147]
JC virus	Yeast	RNAi	IL 10	Purified JC-VLPs were mixed with shRNA in a capsid buffer (150 mM NaCl, 10 mM Tris-HCl, 10 mM CaCl_2_) then diluted with distilled water (an osmotic shock)	[148]
HPV	Sf21 /HEK 293 cell	DNA plasmid (up to 8 kb) Interaction between capsid protein and dsDNA	-	VLP disassembly in the presence of DTT (EGTA)/Reassembly and DNA packaging in the buffer containing CaCl_2_, (ATP)	[37,38,39]
CCMV	Plant	ssDNA, dsDNA Electrostatic interaction to capsid protein	-	VLP disassembly in 5× assembly buffer (250 mM Tris-HCl containing 250 mM NaCl, 50 mM KCl, 25 mM MgCl_2_, pH 7.2)/Reassembly and DNA packaging by adding the mixture of capsid protein and DNA to 1× assembly buffer	[54]
CCMV	Plant/*E. coli*	siRNA, mRNA, Enzyme (HRP) Electrostatic interaction to capsid protein	FOXA1 using siRNA	VLP disassembly in the high salt concentration, neutral pH buffer/Reassembly and cargoes packaging by dialyzing to the first assembly buffer (50 mM Tris pH 7.2, 50 mM NaCl, 10 mM KCl, 5 mM MgCl_2_, 1 mM DTT), then the second buffer (50 mM NaCH_3_COO, 8 mM Mg(CH_3_COO)_2_, pH 4.5)	[31,32,33]
CMV	Plant	DNA, protein, fluorophore Electrostatic interaction with capsid protein	-	VLP disassembly by LiCl/Reassembly and packaging by dialyzing against an assembly buffer (20 mM Tris-HCl, 80 mM KCl, 1 mM DTT, 1 mM MgCl_2_, pH 7.2)	[149]
CCMV	*E. coli*	DOX Conjugated to capsid proteins	Cancer cells using folic acid	Purified proteins were dialyzed to the buffer of 0.1 M NaCH_3_COO, 0.1 M NaCl, pH 4.8	[150]
MPyV	*E. coli*	GFP, m-Ruby3 protein Genetic fusion to capsid protein	-	Purified proteins-linked capsomeres were dialyzed to 20 mM Tris, 0.5 M (NH_4_)_2_SO_4_, 1 mM CaCl_2_, 5% glycerol buffer	[75]
RV	*E. coli*	DOX Conjugated to capsid protein	Hepatoma cells using lactobionic acid	Purified, denatured capsid proteins (in urea) were dialyzed to CH_3_COOH/CH_3_COONa buffer at pH 4.5	[21]
SV40	Sf9	Magnetic nano-particles (MNPs) Electrostatic interaction between capsid protein and MNPs	EGF receptor	VP1 capsid proteins of SV40 were produced in Sf9 cells, pentamers were purified and assembled around magnetic nano-particles in MOPS buffer (20 mM MOPS-NaOH, 150 mM NaCl, 2 mM CaCl_2_, pH 7.0)	[151]
SV40	*E. coli*	Quantum dots (QDs) Direct association between His-tag in VP1 protein and Zn^2+^ on the QDs surface	-	VP1 capsid proteins with a His-tag were produced in *E. coli*, purified VP1 pentamers and QDs were mixed and dialyzed against an assembly buffer (10 mM Tris-HCl, 1 mM CaCl_2_, 250 mM NaCl, 5% glycerol, pH 7.2)	[152]
BMV	Plant	Gold nanoparticles Electrostatic interaction with capsid protein	-	Purified BMV proteins and gold particles were mixed in TKM buffer (10 mM Tris-HCl, 1 M KCl, 5 mM MgCl_2_, pH 7.4 and dialyzed against an assembly buffer (50 mM Tris-HCl, 50 mM NaCl, 10 mM KCl, 5 mM MgCl_2_, pH 7.4), then against SAMA buffer (50 mM NaOAc, 8 mM Mg(OAc)_2_, pH 4.5)	[153]
RRV	*E. coli*	Gold nanoparticles Electrostatic interaction with capsid protein	-	Purified capsid proteins and functionalized GNPs were mixed and dialyzed against an assembly buffer (20 mM Tris-HCl, 50 mM NaCl, 10 mM KCl, 5 mM MgCl_2_, pH 7.4)	[154]

^a^ BMV: Brome mosaic virus; CCMV: Cowpea chlorotic mottle virus; CMV: cucumber mosaic virus; HBc: Hepatitis B core; JC: John Cunningham virus/human polyomavirus 2; MPyV: Murine polyomavirus; MS2: Bacteriophage MS2; P22: Bacteriophage P22; Qβ: Bacteriophage Qβ; RRV: Ross River virus; RV: Rotavirus; SV40: Simian virus 40; DOX: doxycycline; EDTA: ethylenediaminetetraacetic acid.

## Data Availability

Not applicable.

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
