# Peer review of "In Vitro Assembly of Virus-Like Particles and Their Applications"

_life, 2021, doi:10.3390/life11040334_

Round 1
Reviewer 1 Report
The author set their focus on the cell-free assembly of virus-like particles and their application on this review. The topic is very attracting due the global Covid-19 pandemic. However, the content of this review was not engaging. In particular, the structure of this review lacked a logic narrative. Different paragraphs seemed just loosely assembled together. It was really hard to find what was the exact point the author would like to bring the reader. At last, I would be very careful to use the term “cell-free”, because in the synthetic biology community this does not equal to the application or assemble using purified protein. Indeed, there were also merging cell-free protein synthesis technique for virus particles assembly (Tinafar et al. BMC Biology (2019) 17:64). Here are several minor points: Line 270 Table 1. Please give a period for the data you presented here as well as all the data sources. Line 325 Table 2 The same information needed as indicated above. Line 347-362 Was this relevant to this manuscript? If not, please delete it.Author Response
The author set their focus on the cell-free assembly of virus-like particles and their application on this review. The topic is very attracting due the global Covid-19 pandemic. However, the content of this review was not engaging. In particular, the structure of this review lacked a logic narrative. Different paragraphs seemed just loosely assembled together. It was really hard to find what was the exact point the author would like to bring the reader. At last, I would be very careful to use the term “cell-free”, because in the synthetic biology community this does not equal to the application or assemble using purified protein. Indeed, there were also merging cell-free protein synthesis technique for virus particles assembly (Tinafar et al. BMC Biology (2019) 17:64).
Answer: We appreciate the comment. We wrote new paragraphs in the ‘‘Introduction’’ section (paragraph 1 and 4) to clearly define the scope of the review.
We changed the title (from ‘‘cell-free’’ to ‘‘In vitro’’ assembly of virus-like particles and their applications) to clarify the statement.
We provided more information about cell-free protein synthesis technique for virus particle assembly (section 2.1, paragraph 2; section 3.1, last paragraph; Table 1) and cited the suggested reference.
Here are several minor points:
Line 270 Table 1. Please give a period for the data you presented here as well as all the data sources. Line 325 Table 2 The same information needed as indicated above.
Answer: We provided all the data sources (column ‘‘Ref.’’ in Table 1 and 2). The period of time for the data can be found in the added sources.
Line 347-362 Was this relevant to this manuscript? If not, please delete it.
Answer: We deleted it. This was not in our text but added by the publishing system.
Reviewer 2 Report
Major comments:
- Line 98. The word “evaluated” derives from the word “evaluate”, meaning to assess or examine something, and is not properly used in the sentence. What information do you want to pass about the effect of lower temperature in the subunit exchange with assembled capsid shells?
- Lines 250-60. VLPs should mimic the structure of authentic virus, including size and organization. Vaccines based on S-trimers are not VLP-based vaccines, but rather subunit vaccines. Please consider excluding this paragraph, and the rows of Table 1 addressing these examples.
- In the manuscript, more attention is given to cell-free assembly of VLPs, with monomers derived from capsid disassembly of VLPs produced inside living cells, e. the host cells. VLP assembly using protein subunits produced from crude cell extracts, that supply all the necessary elements for the recombinant expression of proteins (cell-free protein synthesis), is also a promising starting point. The manuscript could benefit from a revision on cell-free assembly of VLPs with proteins produced using the CFPS system.
- Additionally, the title is misleading, since “Cell‑free Assembly…” can also mean that both production and assembly are performed without the use of live cells. But in fact, the manuscript focuses on the production of VLPs in living cells, followed by disassembly and re-assembly in vitro. Hence, a more accurate title could be “In vitro assembly of virus-like particles and their applications”.
- In any review tackling a specific topic, it is relevant to discuss what still remains to be done and future prospects. I think the conclusion section should be more elaborated in this regard.
Minor comments:
- Line 43. There is a typo in “protiens”.
- Line 79. By using “since its effect” the phrase is incomplete, requiring an outcome. The words “since its effect” should be altered to “since it affects”. Alternatively, the word “since” should be changed to “due to” or “because”, or similar. In addition, the word “on” that precedes “the capsid-protein charge” should be deleted.
- Line 85. There is an extra space between the words “that” and “an”.
- Line 138. There is no reference to Figure 1 in the text of the manuscript.
- Lines 163-5. Unnecessary repetition when referring to the same subject, VLPs. I suggest rephrasing the sentence as: “A VLP derived from a pathogenic virus may be used to elicit an immune response directly to the parental virus, or function as a scaffold to present epitopes.”
- Line 177. Human papillomavirus (HPV) is in the third person of singular so it should be written as “associates”.
- Line 222. I guess that the word “there” should be “three”.
- Line 262-3. The word “presenting” should be used instead of “presented”.
- Suggestion: Tables 1 and 2 probably could fit in a single page in a landscape orientation; this would allow to improve the format of the “Assembly method” column.
Author Response
Major comments:
Line 98. The word “evaluated” derives from the word “evaluate”, meaning to assess or examine something, and is not properly used in the sentence. What information do you want to pass about the effect of lower temperature in the subunit exchange with assembled capsid shells?
Answer: We corrected this mistake (changed ‘‘evaluated’’ to “elevated”).
Lines 250-60. VLPs should mimic the structure of authentic virus, including size and organization. Vaccines based on S-trimers are not VLP-based vaccines, but rather subunit vaccines. Please consider excluding this paragraph, and the rows of Table 1 addressing these examples.
Answer: We agree with the reviewer that VLPs generally mimic the structure of authentic virus. The widely accepted definition of VLPs is as highly organized supramolecular multiprotein nanoparticles (ranging from 20 to 200 nm) and the the size and organization can be different from the parental viruses. The reviewer is right that the mentioned vaccines are S-trimers, and could therefore be defined as subunit vaccines. Since the S-trimers are anchored within a polysorbate 80 micelle to form a protein nanoparticle of about 27 nm, the vaccines can also be defined as VLPs to distinguish from typical subunit vaccines, which is often used in the literature.
We clarified the statement in section 3.1, paragraph 9. We also wrote additional sentences in the ‘‘introduction’’ section, paragraph 1 to discuss about the VLP definition.
In the manuscript, more attention is given to cell-free assembly of VLPs, with monomers derived from capsid disassembly of VLPs produced inside living cells, e. the host cells. VLP assembly using protein subunits produced from crude cell extracts, that supply all the necessary elements for the recombinant expression of proteins (cell-free protein synthesis), is also a promising starting point. The manuscript could benefit from a revision on cell-free assembly of VLPs with proteins produced using the CFPS system.
Answer: We appreciate the review comment. We added the CFPS system and relevant references (section 3.1, last paragraph). The example of the CFPS system application for a Hepatitis B vaccine candidate was also added in Table 1.
Additionally, the title is misleading, since “Cell‑free Assembly…” can also mean that both production and assembly are performed without the use of live cells. But in fact, the manuscript focuses on the production of VLPs in living cells, followed by disassembly and re-assembly in vitro. Hence, a more accurate title could be “In vitro assembly of virus-like particles and their applications”.
Answer: We changed the title as suggested (In vitro assembly of virus-like particles and their applications).
In any review tackling a specific topic, it is relevant to discuss what still remains to be done and future prospects. I think the conclusion section should be more elaborated in this regard.
Answer: We added a new paragraph in the ‘‘conclusion’’ section (paragraph 2).
Minor comments:
Line 43. There is a typo in “protiens”.
Answer: We corrected this typo.
Line 79. By using “since its effect” the phrase is incomplete, requiring an outcome. The words “since its effect” should be altered to “since it affects”. Alternatively, the word “since” should be changed to “due to” or “because”, or similar. In addition, the word “on” that precedes “the capsid-protein charge” should be deleted.
Answer: We rephrased as suggested.
Line 85. There is an extra space between the words “that” and “an”.
Answer: We corrected this typo.
Line 138. There is no reference to Figure 1 in the text of the manuscript.
Answer: We added ‘‘Figure 1’’ in the text (section 2.2, paragraph 1).
Lines 163-5. Unnecessary repetition when referring to the same subject, VLPs. I suggest rephrasing the sentence as: “A VLP derived from a pathogenic virus may be used to elicit an immune response directly to the parental virus, or function as a scaffold to present epitopes.”
Answer: We rephrased the sentence as suggested (section 3.1, paragraph 1).
Line 177. Human papillomavirus (HPV) is in the third person of singular so it should be written as “associates”.
Answer: We corrected this error.
Line 222. I guess that the word “there” should be “three”.
Answer: We corrected this typo.
Line 262-3. The word “presenting” should be used instead of “presented”.
Answer: We corrected this mistake.
Suggestion: Tables 1 and 2 probably could fit in a single page in a landscape orientation; this would allow to improve the format of the “Assembly method” column.
Answer: We appreciate the suggestion and will recommend this format to the journal editor.
Reviewer 3 Report
In the manuscript entitled: “Cell-free assembly of virus-like particles and their applications”, the authors provided a general overview on cell-free capsid assembly of VLPs and their applications in vaccine development and cargo delivery. In the first section of the review, they described the expression host systems employed to produce viral capsid proteins, factors affecting the in vitro assembly and methods for VLP characterization. Then, they focused on the different in-vitro packaging strategies pursued to purify and develop VLP-based vaccine/delivery platforms already licensed or under preclinical/clinical evaluation.
This manuscript would be of interest to the VLP research field. Providing valuable alternative strategies to the traditional methods would overall save time, cut costs and implement the development of VLP-based platforms for biomedical applications. The manuscript is concisely written. However, some sections lack of relevant details and thus is not suitable for publication in its current form.
I would appreciate if the authors could take into account the following comments and suggestions.
Major Comments
- Introduction. VLPs are a class of nanoparticle delivery systems that comprise a variety of nano-scale size materials, like solid nanoparticles and emulsions. They can be either enveloped or non-enveloped, chimeric or conjugated, and harbor intrinsic features as the particulate structure. Currently, several native viruses have been considered and mimicked by VLPs. I would suggest to briefly introduce and clarify these concepts in the “Introduction” section to make the content of the manuscript even more comprehensive to non-VLP researchers.
- Section 2.1. Expression systems for VLP production should be clearly described and presented as mammalian, insect, plant, yeast, bacteria, and cell-free protein synthesis (CFPS) systems, with highlighted pros and/or cons. Among factors that could increase solubility, assist assembly and folding of proteins, I would also discuss: codon optimization strategies, expression vectors containing rare tRNA and molecular chaperones.
- Section 2.2.2. Nucleic acids were mentioned among factors affecting the in-vitro assembly of VLPs. The authors should also discuss the potential effect of encapsidated DNA/RNA on triggering innate immune responses.
- Section 2.3. Among assays for VLP characterization I would also mention: Bradford protein assay; Western Blot; assays to detect protein-nucleic acid complexes or to investigate the glycosylation pattern; mass-spectrometry for VLP composition.
- Section 3. The immunological mechanism of action of VLPs should be explained and schematically represented by a figure (likely Figure 2). That might be helpful to the readers.
- Section 3.1. As result of expression in E. coli, VLPs might be contaminated by LPS. Additional informations regarding strategies pursued to lower endotoxin levels in in-vitro assembled VLPs would be of interest to researchers working with this system.
- Table 1. The authors stated: “A list of developed and approved vaccine candidates is presented in Table 1” (line 176). To my surprise, few preclinical studies have been listed. Concerning SARS-CoV-2 (e.g.), there are several vaccine candidates exploiting the VLP platform that are under preclinical evaluation and more than one in clinical phase. They authors should clarify how they did retrieved the listed studies, they should update the current table, also providing the registered trial number (when possible). However, I would present these studies in two separate tables, with Table 1 listing the VLP-based vaccines commercially available and Table 2 listing the candidates under preclinical and clinical evaluation. In the headings of both tables I would also include the “virus” column.
- The in-vitro manufacturing process requires additional steps. The authors should also discuss issues related to manufacturing costs, stability and yield of in-vitro assembled VLPs.
Minor Comments
- Page 1, line 21. The statement “virus’s transduction potential” is confusing. Please, reword for clarity;
- Page 1, lines 29-31. The authors stated: “For the in-vitro assembly, the capsid proteins are expressed and purified (how? from?), or the proteins are obtained from in-vivo VLPs via a disassembly procedure …”. Is there something missing in this sentence? Please, reword for clarity;
- Page 1, line 33. Please delete comma in between “conditions” and “which”;
- Page 1, lines 35-36. I would state: “different viral genotypes” rather than “different diseases”, since ref. #6 and #7 refer to a bi- or nine-valent HPV vaccine;
- Page 1, line 38. Please add a reference at the end of the sentence;
- Page 1, lines 42 and page 2, line 65. Would “obtained” be better than “acquired”?;
- Page 1, lines 43. Correct “protiens” in “proteins”;
- Please, mention Figure 1 in the body text of the manuscript, providing also a brief figure legend.
- Page 2, line 72. Please, check the heading of subsection 2.2. Would “Factors impacting on in vitro assembly” be better?;
- Page 2, line 91. Please, clarify what does “phase diagrams” state for;
- Page 4, line 157. Please, clarify what does “intact-particle” state for;
- Page 4, line 159. Please, check the heading of section 3. Would “Applications of in vitro-assembled VLPs” be better?;
- Please, provide URL for references #72, #85 and #86;
- Page 6, line 222. Correct “there” in “three”;
- Page 6, lines 228-229 and lines 261-262. Would “under preclinical evaluation” be better than “preclinical trial/s”?;
- Page 6. Sentence from line 247 (Hence …) to line 249 is confusing. Please, reword for clarity;
- Page 8, line 295. Correct “was achieved” in “ were achieved”.
Author Response
Comments and Suggestions for Authors
In the manuscript entitled: “Cell-free assembly of virus-like particles and their applications”, the authors provided a general overview on cell-free capsid assembly of VLPs and their applications in vaccine development and cargo delivery. In the first section of the review, they described the expression host systems employed to produce viral capsid proteins, factors affecting the in vitro assembly and methods for VLP characterization. Then, they focused on the different in-vitro packaging strategies pursued to purify and develop VLP-based vaccine/delivery platforms already licensed or under preclinical/clinical evaluation.
This manuscript would be of interest to the VLP research field. Providing valuable alternative strategies to the traditional methods would overall save time, cut costs and implement the development of VLP-based platforms for biomedical applications. The manuscript is concisely written. However, some sections lack of relevant details and thus is not suitable for publication in its current form.
I would appreciate if the authors could take into account the following comments and suggestions.
Major Comments
Introduction. VLPs are a class of nanoparticle delivery systems that comprise a variety of nano-scale size materials, like solid nanoparticles and emulsions. They can be either enveloped or non-enveloped, chimeric or conjugated, and harbor intrinsic features as the particulate structure. Currently, several native viruses have been considered and mimicked by VLPs. I would suggest to briefly introduce and clarify these concepts in the “Introduction” section to make the content of the manuscript even more comprehensive to non-VLP researchers.
Answer: We added the suggested information of VLPs in the ‘‘Introduction’’ section (paragraph 1) to clarify the concept. We added relevant references:
Jeevanandam, J.; Barhoum, A.; Chan, Y.S.; Dufresne, A.; Danquah, M.K. Review on nanoparticles and nanostructured materials: history, sources, toxicity and regulations. Beilstein J. Nanotechnol. 2018, 9, 1050–1074.
Lua, L.H.L.; Connors, N.K.; Sainsbury, F.; Chuan, Y.P.; Wibowo, N.; Middelberg, A.P.J. Bioengineering virus-like particles as vaccines. Biotechnol. Bioeng. 2014, 111, 425–440.
Section 2.1. Expression systems for VLP production should be clearly described and presented as mammalian, insect, plant, yeast, bacteria, and cell-free protein synthesis (CFPS) systems, with highlighted pros and/or cons. Among factors that could increase solubility, assist assembly and folding of proteins, I would also discuss: codon optimization strategies, expression vectors containing rare tRNA and molecular chaperones.
Answer: We added the pros of CFPS and respective references (Paragraph 2).
We also added information on codon optimization and expression vectors as well as relevant references as suggested (Paragraph 1).
Section 2.2.2. Nucleic acids were mentioned among factors affecting the in-vitro assembly of VLPs. The authors should also discuss the potential effect of encapsidated DNA/RNA on triggering innate immune responses.
Answer: We added the potential effect of encapsidated DNA/RNA on immune responses and provided a reference (paragraph 1).
Section 2.3. Among assays for VLP characterization I would also mention: Bradford protein assay; Western Blot; assays to detect protein-nucleic acid complexes or to investigate the glycosylation pattern; mass-spectrometry for VLP composition.
Answer: We added the techniques Bradford protein assay, Western Blot, gel retardation assay, optical tweezers (OT) and acoustic force spectroscopy (AFS) to determine protein-nucleic acid interaction, and mass spectrometry. We also wrote the additional paragraph to discuss different methods for VLP assembly pathway characterization (paragraph 2) and cited the relevant references.
Section 3. The immunological mechanism of action of VLPs should be explained and schematically represented by a figure (likely Figure 2). That might be helpful to the readers.
Answer: We appreciate the reviewer comment. We added two review references that discuss in detail the interaction between VLPs and the immune system and clarified the statement in section 3.1, paragraph 1. References:
Mohsen, M.; Gomes, A.; Vogel, M.; Bachmann, M. Interaction of Viral Capsid-Derived Virus-Like Particles (VLPs) with the Innate Immune System. Vaccines 2018, 6, 37.
Jennings, G.T.; Bachmann, M.F. The coming of age of virus-like particle vaccines. Biol. Chem. 2008, 389, 521–536.
We agree that the new figure to explain the immunological mechanism might be helpful, however we think that it is beyond the scope of this review that focuses on in vitro assembly of VLPs.
Section 3.1. As result of expression in E. coli, VLPs might be contaminated by LPS. Additional informations regarding strategies pursued to lower endotoxin levels in in-vitro assembled VLPs would be of interest to researchers working with this system.
Answer: We added additional information and reference as suggested (section 3.1, paragraph 6).
Schädlich, L.; Senger, T.; Kirschning, C.J.; Müller, M.; Gissmann, L. Refining HPV 16 L1 purification from E. coli: Reducing endotoxin contaminations and their impact on immunogenicity. Vaccine 2009, 27, 1511–1522.
Table 1. The authors stated: “A list of developed and approved vaccine candidates is presented in Table 1” (line 176). To my surprise, few preclinical studies have been listed. Concerning SARS-CoV-2 (e.g.), there are several vaccine candidates exploiting the VLP platform that are under preclinical evaluation and more than one in clinical phase. The authors should clarify how they did retrieved the listed studies, they should update the current table, also providing the registered trial number (when possible). However, I would present these studies in two separate tables, with Table 1 listing the VLP-based vaccines commercially available and Table 2 listing the candidates under preclinical and clinical evaluation. In the headings of both tables I would also include the “virus” column.
Answer: We appreciate the comment. We agree with the reviewer that several vaccine candidates are exploiting the VLP platform under preclinical evaluation and in clinical (e.g. SARS-CoV-2 vaccine). However, most of VLP vaccine candidates for SARS-CoV-2 have been produced using the in vivo method, not by in-vitro assembly. In this table, we only listed in-vitro assembly vaccine candidates, which were assembled into VLPs outside living cells. Therefore, for SARS-CoV-2 vaccines, we mentioned the in-vitro VLP vaccine candidate produced by the Novavax company.
We provided the registered trial number in the ‘‘development stage’’ column, Table 1 as suggested.
In the table, we want to present different methods to produce VLPs of each vaccine candidate (both in preclinical evaluation and clinical trials) to highlight the method differences. We think that it would be better for readers to compare when summarizing in one table.
We added the ‘‘virus name’’ into the ‘‘vaccine antigen’’ column in the table as suggested.
We also added new VLP candidates for therapeutic delivery in section 3.2 as well as Table 2.
The in-vitro manufacturing process requires additional steps. The authors should also discuss issues related to manufacturing costs, stability and yield of in-vitro assembled VLPs.
Answer: We added the related manufacturing factors in section 3.1, paragraph 6 and 8.
Minor Comments
Page 1, line 21. The statement “virus’s transduction potential” is confusing. Please, reword for clarity;
Answer: We rephrased to clarify the statement.
Page 1, lines 29-31. The authors stated: “For the in-vitro assembly, the capsid proteins are expressed and purified (how? from?), or the proteins are obtained from in-vivo VLPs via a disassembly procedure …”. Is there something missing in this sentence? Please, reword for clarity;
Answer: We added a new phrase to clarify the sentence.
Page 1, line 33. Please delete comma in between “conditions” and “which”;
Answer: We assumed that the information provided can be seen as restrictive or nonrestrictive; we deleted the comma.
Page 1, lines 35-36. I would state: “different viral genotypes” rather than “different diseases”, since ref. #6 and #7 refer to a bi- or nine-valent HPV vaccine;
Answer: We changed the word as suggested.
Page 1, line 38. Please add a reference at the end of the sentence;
Answer: We added the reference.
Page 1, lines 42 and page 2, line 65. Would “obtained” be better than “acquired”?;
Answer: We changed the word as suggested.
Page 1, lines 43. Correct “protiens” in “proteins”;
Answer: We corrected this typo.
Please, mention Figure 1 in the body text of the manuscript, providing also a brief figure legend.
Answer: We mentioned Figure 1 in the text and provided the figure legend as suggested.
Page 2, line 72. Please, check the heading of subsection 2.2. Would “Factors impacting on in vitro assembly” be better?;
Answer: We changed the heading as suggested.
Page 2, line 91. Please, clarify what does “phase diagrams” state for;
Answer: We clarified the sentence and provided additional information about capsid assembly in section 2.2 (paragraph 1) to clarify phase diagrams of VLP formation.
Page 4, line 157. Please, clarify what does “intact-particle” state for;
Answer: We changed the word to clarify the sentence.
Page 4, line 159. Please, check the heading of section 3. Would “Applications of in vitro-assembled VLPs” be better?;
Answer: We changed the heading as suggested.
Please, provide URL for references #72, #85 and #86;
Answer: We provided the URLs.
Page 6, line 222. Correct “there” in “three”;
Answer: We corrected the typo.
Page 6, lines 228-229 and lines 261-262. Would “under preclinical evaluation” be better than “preclinical trial/s”?;
Answer: We changed the phrase in the mentioned sentence and also in the tables as suggested.
Page 6. Sentence from line 247 (Hence …) to line 249 is confusing. Please, reword for clarity;
Answer: We rephrased the sentence.
Page 8, line 295. Correct “was achieved” in “were achieved”.
Answer: We rephrased the sentence.
Reviewer 4 Report
This is a review of the use of in vitro assembly of VLPs for the purpose of vaccines and drug delivery. The authors give a reasonable though slightly superficial review of the field, but there are 3 primary concerns.
- The authors need to more clearly define the scope of their review in the beginning of the review--ideally in the introduction. There are quite a few reviews of VLPs, so it would be helpful to point the readers to other reviews that are useful, but then clearly identify what contribution/area this specific review will cover. It appears that the review is largely focused on the specifics of methodology for VLP assembly in vitro. This should be clearly stated so the reader is aware of the specific scope when they start reading.
- The review needs to provide some context and analysis. For instance--giving pros and cons of the specific methodologies presented. Throughout, the review would benefit from the unique perspectives of the authors.
- Finally, the authors need to provide a forward-looking conclusion. What do the authors think is the next steps, questions, or areas that need to be explored in the field?
Author Response
This is a review of the use of in vitro assembly of VLPs for the purpose of vaccines and drug delivery. The authors give a reasonable though slightly superficial review of the field, but there are 3 primary concerns.
The authors need to more clearly define the scope of their review in the beginning of the review--ideally in the introduction. There are quite a few reviews of VLPs, so it would be helpful to point the readers to other reviews that are useful, but then clearly identify what contribution/area this specific review will cover. It appears that the review is largely focused on the specifics of methodology for VLP assembly in vitro. This should be clearly stated so the reader is aware of the specific scope when they start reading.
Answer: We appreciate the comment. We wrote a new paragraph in the ‘‘introduction’’ section (paragraph 4) to clearly define the scope of this review and cited relevant references.
The review needs to provide some context and analysis. For instance--giving pros and cons of the specific methodologies presented. Throughout, the review would benefit from the unique perspectives of the authors.
Answer: We discussed more about the expression of capsid protein using protein cell-free synthesis (section 2.1), the effect of nucleic acid on VLP assembly application (section 2.2.2). We also added more information about in vitro VLP manufacturing (section 3.1).
Finally, the authors need to provide a forward-looking conclusion. What do the authors think is the next steps, questions, or areas that need to be explored in the field?
Answer: We provided a forward-looking conclusion (the ‘‘conclusion’’ section, paragraph 2)
Round 2
Reviewer 2 Report
Minor comments
Line 23. Double "be" at the end of the sentence.
Line 37. A comma should be added after the word "impurities".
Line 198. Since the sentence refers to the future, "will enable" probably suits better than "enables".
Line 403. Double words "clinic clinical".
Author Response
We applied the suggested changes.
Reviewer 3 Report
The authors addressed all the comments and overall improved the quality of the manuscript. I would endorse its publication.
Only one minor comment: Please, correctly cite in the body text of the manuscript (at line 203) refs. #6 and #101 citing the first authors’ name.
Author Response
We applied the suggested changes.